# Teledermatology versus Face-to-Face Dermatology: An Analysis of Cost-Effectiveness from Eight Studies from Europe and the United States

**DOI:** 10.3390/ijerph19052534

**Published:** 2022-02-22

**Authors:** Remedios López-Liria, María Ángeles Valverde-Martínez, Antonio López-Villegas, Rafael Jesús Bautista-Mesa, Francisco Antonio Vega-Ramírez, Salvador Peiró, Cesar Leal-Costa

**Affiliations:** 1Health Research Centre, Department of Nursing, Physiotherapy and Medicine, University of Almería, Carretera del Sacramento s/n, La Cañada de San Urbano, 04120 Almeria, Spain; rll040@ual.es (R.L.-L.); mvm637@ual.es (M.Á.V.-M.); 2Social Involvement of Critical and Emergency Medicine, CTS-609 Research Group, Poniente Hospital, 04700 El Ejido, Spain; 3Economic-Financial Directorate, Alto Guadalquivir Health Agency, 23740 Andujar, Spain; rafael.bautista.mesa@gmail.com; 4Hum-498 Research Team, University of Almeria, 04120 Almeria, Spain; franavega@hotmail.com; 5Health Services Research Unit, FISABIO-Public Health, 46020 Valencia, Spain; peiro_bor@gva.es; 6Nursing Department, University of Murcia, 30120 El Palmar, Spain; cleal@um.es

**Keywords:** cost–benefit analysis, follow-up studies, health-related quality of life, pacemakers, teledermatology, telemedicine

## Abstract

(1) Background: The aim of this systematic review was to compare the cost-effectiveness of two follow-up methods (face-to-face and telemedicine) used in dermatology in the last ten years. (2) Methods: A search for articles that included economic analyses was conducted in August 2021 in the databases PubMed, Medline, Scielo and Scopus using the following keywords: “Cost–Benefit Analysis”, “Dermatology”, “Telemedicine”, “Primary Health Care”, as well as other search terms and following the PICOS eligibility criteria. (3) Results: Three clinical trials and five observational studies were analyzed, providing information for approximately 16,539 patients (including four cost-minimization or saving analyses, three cost-effectiveness analyses, and one cost–utility analysis) in Europe and the United States. They describe the follow-up procedures in each of the cases and measure and analyze the direct and indirect costs and effectiveness. All the articles indicate that teledermatology lowers costs and proves satisfactory to both patients and professionals. (4) Conclusions: Although it has been found that follow-up via teledermatology can be more efficient than traditional hospital follow-up, more work is needed to establish evaluation protocols and procedures that measure key variables more equally and demonstrate the quality of the evidence of said studies.

## 1. Introduction

Telemedicine is proving to be an efficient tool for improving remote medical assistance [1]. Dermatology is the clinical speciality best suited for telemedicine, owing to the visibility of dermatological conditions [2,3,4].

The concept of teledermatology (TD) was first introduced in 1995 to offer remote dermatology services [5]. More recently, the arrival of the COVID-19 pandemic changed the scheme of in-person consultation within the healthcare system, except for those cases deemed emergencies [3,5]. Although TD was already in use worldwide, the arrival of COVID-19 and the subsequent confinement period consolidated the application of the tool [4,6]. Prior to this time, it had primarily been a resource aimed at rural populations with less convenient accessibility [7].

Dermatological conditions are a frequent reason for primary care (PC) visits [8]. In general terms, TD is an easy-access tool for professionals in this field worldwide [6,9]. In some countries, TD service begins with PC physicians/nurses taking a photograph of the lesion in question and attaching it to the patient’s electronic medical records, along with a brief clinical description. Subsequently, at the referral hospital, the consultant dermatologists view the electronic medical records, observe the images and recommend a treatment or plan of action. Later, PC physicians evaluate these recommendations and contact the patient to convey the results [8,10].

There are several fields in TD care: synchronous, asynchronous or the combination of both [2,11]. Synchronous TD offers immediate diagnosis, as the dermatologist views the photograph of the patient in real time (live). In the asynchronous method, or storage TD, the images of the patients are loaded to a platform using a PC and sent to a dermatologist who later establishes a diagnosis. In clinical practice, the most widely utilized is the asynchronous model [8,12]. Those patients with an inconclusive diagnosis or who require an in-person appointment are contacted more quickly, thereby reducing any waiting lists.

TD offers certain advantages over traditional dermatology consultations. It reduces waiting times and costs, improves access in rural areas and/or for patients who experience difficulties in attending in-person appointments and decreases appointment cancellations and absenteeism. Moreover, TD proves to be a useful diagnostic and follow-up tool, favoring early referral in emergency cases [11,13,14]. There are, however, other aspects that adversely affect TD. These include technological barriers (poor internet connection, limited access to platforms, poor image quality) or lack of technological skills among professionals and patients, which could hinder attention to geriatric patients or those with a different language, ultimately resulting in the making of possibly erroneous diagnoses [14,15].

The use of TD is greater in countries in North America and Europe [9]. Spain is one of the leading countries with active TD programs [6]. However, in other countries, such as Argentina, TD is currently in its early stages of development [4]. Approximately 38% of countries worldwide have integrated TD programs. In Australia, Holland and some parts of the United States, TD represents a key part of the healthcare system [16,17], as many rural areas lack dermatologists, thereby favoring its utilization [17].

It seems that the development and expansion of TD is having a positive influence in terms of cost-effectiveness, a fact supported by the few studies and reviews found of this recent practice [4,9,18]. However, there are even fewer investigations that focus on the analysis of the cost-effectiveness of TD compared to traditional consultations. It would be rather interesting to establish a comparative between both treatment methods and extract results that could be extrapolated and applied to improve patient care. Bearing this in mind, the aim of this systematic review is to compare the cost-effectiveness of two follow-up methods (face-to-face and telemedicine) used in dermatology over the last ten years.

## 2. Materials and Methods

In August 2021, a systematic review was conducted following the standards set out in the PRISMA statement for systematic reviews [19,20]. Said review was previously registered with the International Prospective Register of systematic reviews (PROSPERO) and was assigned number CRD42021267213. A meta-analysis could not be carried out due to the range of study designs used and inconsistent outcome reporting.

A search was conducted for articles published in the last 10 years in PubMed, Medline, Scielo and Scopus. The target population was users of dermatology services and the material was gathered from the studies present in the databases mentioned, existing gray literature and Web of Science. Additionally, manual searches were undertaken to locate bibliographic references deemed to be of interest—those included in systematic reviews and prior meta-analyses. The inclusion criteria were:Articles containing the MeSH terms: “Cost–Benefit Analysis”, “Dermatology”, “Telemedicine”, “Primary Health Care”. Other search terms included: “Cost–Utility Analysis”, “Economic Evaluation”, and “Cost-Effectiveness Analysis”, “Teledermatology”, “Skin Disease”, “Telehealth”, “Remote Consult” (conventional or “face-to-face” or standard or in-person).Clinical trials or observational studies comparing the two follow-up methods (face-to-face versus TD).

The exclusion criteria were:Description of a single method for intervention or follow-up, without comparison.Clinical guidelines, systematic reviews and meta-analyses.

The search strategy in the different databases is described in Table 1.

The PICOS eligibility criteria (participants, intervention, comparison intervention, results and study design) were used for article selection: participants would be users of dermatology services. Intervention would involve the application of face-to-face dermatological monitoring follow-up (in-person at a hospital) or remote monitoring follow-up using telemedicine (teledermatology). The results measured were the costs of these follow-up methods for dermatological conditions, as well as secondary measures, such as the effectiveness or satisfaction of users and professionals. As for the type of study required, investigations had to be observational studies or clinical trials.

Articles should evaluate (main variables): number of visits, economic impact or identification of costs, such as those directly attributable to TD and/or face-to-face dermatology service (including cameras, hardware and staff).

Among the other variables considered were: costs not directly attributable to dermatology service (such as building maintenance, information technology (IT) services, gas, electricity, telephone–internet connections and medical insurance). Also taken into account were costs incurred by patients and society, for example, lost productive time, lost salaries, leisure time lost, time spent traveling to visits and petrol.

Following the application of the search strategy, a total of 71 articles were selected (between 13 and 31 August 2021), which were analyzed by title and abstract.

The extraction and reading of all titles and abstracts of the studies initially selected was carried out independently by two researchers (R.L.-L. and M.Á.V.-M.), who consulted a third individual (A.L.-V.) if there was any disagreement about the inclusion/exclusion of an article.

After an initial screening of the studies considered to be potentially relevant, a full-text critical reading of 27 articles was performed, paying particular attention to the intervention (type of treatment) and cost–benefit evaluation. Finally, it was determined that a total of eight articles met the objective and fulfilled the criteria proposed for this review (Figure 1). A descriptive analysis of the selected results was carried out. Disagreements were resolved through discussion among the reviewers until a consensus was reached.

The quality of the studies was evaluated using the Downs and Black quality assessment method [21]. This method contains 27 items divided into five sections: study quality, external validity, study bias, confounding and selections bias and study power. Scores range from 0 to 28; higher scores indicate a better methodological quality of the study: excellent (26–28), good (20–25), fair (15–19), and poor (<14) This scale has high validity and reliability, ranked as one of the six best quality assessment scales suitable for systematic reviews [22,23].

## 3. Results

Eight articles were included which provided information on 16,539 patients (including four cost-minimization or saving analyses, three cost-effectiveness analyses and one cost–utility analysis]. They described how the follow-ups were conducted for each of the methods and contained at least a cost analysis.

The following Table 2 presents a summary of the studies selected, indicating: type of study, country, number of participants, follow-up method in each group, the variables that measured direct and indirect costs, effectiveness and the main results obtained.

The following section presents the content analysis of the information based on the most important variables (Table 3):

### 3.1. Description of TD Procedure versus Face-to-face Follow-Up

Among the various procedures used in dermatology, all the articles use an asynchonous model, in which the images of the patient are uploaded to a platform along with a brief clinical history, after which the dermatologist establishes a diagnosis and treatment plan [24,25,26,27,28,29,30].

Patients attend an initial consultation with a PC dermatology doctor [24,25,26,27,28,29] or nurse [30] who takes the photos of the affected area and uploads them to an online platform which a dermatologist can access remotely. In the study by Parsi [31], the patients themselves take photos of their own skin and upload them to the platform.

For example, in the study by Zakaria et al. [24], the referring clinicians are required to upload patient photographs and a brief history through a web-based telemedicine platform. An attending dermatologist reviews TD cases weekly and determines which patients require an in-person appointment at the dermatology clinic. When a dermatology visit is not recommended, the referring clinician is expected to coordinate the dermatologist’s recommended workup and treatment plan.

Os-Medendorp [30] describes an e-health portal for patients consisting of e-consultation, a patient-tailored website, monitoring and self-management training.

Some studies consider the preliminary training of the doctor or patient in taking photographs correctly [24,28,30]. Only in the studies by Os-Medendorp et al. [30] and Parsi et al. [31] do patients receive advice on self-care in addition to general information and individualized guidance on their dermatological condition.

In the article by López-Villegas et al. [25], photographs are taken with a dermascope. All others use images taken with a mobile phone or similar devices [24,26,27,28,29,30,31].

The face-to-face or conventional dermatological practice used in all the studies consists of an in-person consultation at a hospital [24,25,26,27,28,29,30,31].

Follow-up periods vary between three [26], six [24] and nine [29] months and one [25,28,30] and five years [27].

### 3.2. Variables That Measure Direct and Indirect Costs

The types of economic analysis and economic perspectives vary greatly among these studies.

In general, nearly all the articles include the labour costs of the heathcare professionals and the patients [24,25,26,29,30,31]. However, in the case of Zarca [28], the salary of patients is not included, as the patients being studied were inmates. The article by Yang [27] includes the costs of PC, dermatology and TD consultations by establishing estimations (minimum, mean and maximum).

Zakaria [24], López-Villegas [25] and Vidal-Alaball [26], assess personal costs (salary and time), technological costs, as well as the number of PC, dermatological and TD consultations. The three studies exclude maintenance costs and those incurred by patients’ companions.

In order to determine the number of face-to-face consultations saved in Spain, López- Villegas [25] and Vidal-Alaball [26] utilize the following equation:Nº TD consultations − Nº consultations requested following

Zakaria [24] utilizes a decision tree model to analyze the difference in costs for patient management and real costs instead of estimations.

López-Villegas [25] and Vidal-Alaball [26] assess, from the patients’ perspective, travel time (according to distance obtained using Google Maps) and the cost of gasoline (km/h, established by the Spanish Ministry of Finance). Both studies assume that the patients use their personal means of transport. They also consider waiting time, consultation time and legal average salary per hour.

### 3.3. Variables That Measure the Effectiveness of TD versus Face-to-Face Follow-Up

A medical treatment is generally considered to be cost-effective based on the following conditions: it provides an added health benefit at an equal or lower cost than the opposing treatment; it provides an added health benefit that is worth an additional cost; or it provides a lesser health benefit but comes with cost savings that are more valuable than the health benefit lost [31].

A variety of questionnaires were used to measure effectiveness. On the one hand, in the study by Yang et al. [27], the PC doctor has to respond to the question: “Without the TD service, how would this patient have otherwise received care for this condition?” with these options: (1)“I would take care of the issue myself”, (2) “I would refer the patient for an in-person dermatologist visit” or (3) “I would refer the patient to urgent care or an emergency room (ER)”. Based on their answers, 60% of cases would be handled by the PC doctor, 35.6% would be referred for a dermatologist visit and 4.4.% to an emergency room. On the other hand, Zarca [28], uses the MAST Model (Model for Assessment of Telemedicine) to conduct a multidimensional evaluation in which PC doctors and dermatologists had to respond to a 6-item survey.

To measure quality of life, Os-Medendorp [30] and Parsi [31] utilize the Dermatology Life Quality Index (DLQI). Os-Medendorp [30] also uses the Infants’ Dermatititis Quality of Life Index (IDQIL), while Parsi [31], applies the “European Quality of Life Survey—5 Dimensions”, along with Quality-Adjusted Life Expectancy (QALE) and Quality-Adjusted Life Year (QALY).

As for the study by Datta et al. [29], quality of life is determined according to the patient’s health condition: living longer with a dermatological condition versus living for a shorter time with perfect health.

For the purpose of gathering information on the status of skin, the “Impact of Chronic Skin Disease on Daily Life” questionnaire was used, along with the VAS scale to measure itch intensity and the “Health and Labour Questionnaire” to determine the loss of productivity [30].

### 3.4. Comparison of Cost-Effectiveness in Each Method of Follow-Up

In the study by Parsi [31], TD proves to have the greatest savings in relation to in-person consultation, which is in keeping with that of Zakaria [24], in which TD obtains a mean savings of $110.12 per patient. The cost reduction, albeit lower, is also found to be evident in the studies by Vidal-Alaball [26] and Datta [29].

The study by López-Villegas [25] indicates that TD effectively saves more than 50% of visits with respect to face-to-face follow-up within the public healthcare system. In the article by Vidal-Alaball [26], only a small percentage of participants required an in-person consultation.

Yang [27] finds that in-person consultations were reduced to 14%, representing significant cost savings. Zarca [28] also identifies an evident cost reduction, yet one that depends on the number of patients treated.

Once again referencing the study by Datta [29], it does not show statistically significant differences between both methods in terms of effectiveness.

Similary, Os-Medendorp [30] fails to indentify any significant differences in terms of the quality of life of patients with atopic dermatitis, finding TD to be as effective as in-person consultation. However, the article by Parsi [31] cites the existence of a slightly higher improvement in the quality of life of patients with the in-person method, yet, in terms of preference, the patients support online treatment.

Regarding satisfaction among doctors, Zarca [28] finds that doctors display greater satisfaction with TD.

### 3.5. Level of Evidence and Quality of Articles Included

Table 4 shows the score of each item of the Downs and Black scale [21]. The mean score of the studies included was 22.12 (range: 17–26), considering that the highest possible score was 28 points. Based on the cut-off points suggested for categorizing studies according to their quality, one article was evaluated as “fair” (15–19 points) and seven were classified as “good” (20–25 points).

## 4. Discussion

This systematic review has described studies that compare an online model (telemedicine) with in-office care in dermatology services. Although both models can be considered highly cost-effective, herein, it has been found that teledermatology reduces costs and is satisfactory to both patients and professionals. Reviews, such as those by Barbieri et al. [32], agree that TD has the potential to provide access to suitable, quality care.

Nonetheless, the implementation of TD requires either investments or improvements in healthcare infrastructures and training for professionals, as well as for patients as relates to prevention [18,33]. From the legislative and ethical point of view, TD is considered to be a care method that should respect patient autonomy. According to European legislation, the informed consent of the patient is only necessary in cases where the images taken reveal the identity of the subject and/or are used for teaching or scientific purposes [10].

Zakaria [24] demonstrated that TD favours the remote classification and treatment of patients with dermatological conditions by means of implementing a triage system, which in turn contributes to a reduction of costs. The study by Barbieri et al. [32] also suggests that TD is effective for classification in dermatological consultations and could increase their efficiency.

Several economic analyses have been performed on traditional TD models; however, most are cost-minimization studies and do not regularly use validated clinical outcomes or standardized cost-effectiveness measures [31,34].

It would be interesting to promote the use of dermascopes to take photographs of patients, as used in the study by López-Villegas [25]. Ferrandiz [35] considers that images taken with a dermascope significantly improve diagnoses when compared to other clinical images. This study highlights the importance of a strong interconnection between PC and specialised care (hospital).

The pandemic and the situation caused by COVID-19 have consolidated the use of TD, with no need to interrupt treatments. What is more, these events favored access to specialized care for individuals who, due to specific circumstances, are unable to attend in person [28] or who live in areas that are distant or difficult to reach [36]. Several studies promote the use of TD in rural or isolated areas, such as Cutler et al. [37], for Haiti; Byamba et al. [1], for Mongolia, one of the least densely populated countries worldwide; and Tran et al. [38], for Cairo, Egypt. Similarly, the review by Kozera et al. [39] supports the use of TD as a valuable service that allows patients in these rural areas to access care.

A systematic review of cost-effectiveness of store-and-forward TD in 2016 [40] suggests that TD can be cost-effective when used as a triage mechanism to reduce face-to-face appointment requirements, although evidence was sparse. It suggests that further economic research is required for TD, which uses dermoscopes in combination with smartphone applications, as well as regarding the possibility and consequences of patients self-capturing and transmitting images.

In the present systematic review, all the studies utilized the method of saving and forwarding images [24,25,26,27,28,29,30,31]. Hadeler et al. [41] states that this type of method is highly satisfactory for the patient, albeit some either declare feeling embarrassed by having photographs taken of their skin or express rejection due to social or religious issues.

All the studies analysed in our revision included pathologies that could be followed-up by TD. Some of them did not specify the dermatological condition [26,27,29]. The rest of the articles included diverse diseases: benign growths [24,25], infection and eczematous dermatitis [24], acnea [28], atopic dermatitis [30], psoriasis [31] or non-benign pathologies (various types of cancer of the skin) [25]. Only one study [25] refers to the cost-effectiveness TD in terms of skin cancer follow-up, although a face-to-face consultation was sometimes necessary.

It would also be necessary to discuss the cost-effectiveness of TD in terms of skin cancer follow-up comparing clinical trials, as during the COVID-19 pandemic this topic or the lack of this follow-up has been of great importance. We have found a study that analyzed the potential benefits gained from the addition of dermoscopic images to an internet-based skin cancer screening system [35]. In this trial, teledermoscopy has rendered high sensitivity and specificity to assist the dermatologist in making referral decisions, consequently improving the proportion of correct decisions where the relevant lesions were skin cancer (melanoma), premalignant lesions (actinic keratosis) or lesions suitable for follow-up (atypical nevus or others) [35].

Finally, all the articles in this review state that both treatment methods are effective [24,25,26,27,28,29,30,31]. All the studies found improvements in quality of life with both online and in-office patients. Studies, such as those by Vyas et al. [42], continue their commitment to TD as being a safe tool with comparable or superior effectiveness to traditional consultation, as well as favoring interprofessional collaboration in time and space. This method eliminates a large number of unnecessary referrals and could be utilized as an education tool for patients.

The limitations of this review resemble those found in the cost-effectiveness studies. They are related to the variability of the direct and indirect costs included or excluded and the sample of patients. It is necessary that the patients have similar characteristics and the sample number not be large so as to obtain more evident and reliable results that can be extrapolated to the rest of the population. In addition to the variety of methods used to evaluate costs, QALY and satisfaction have also been measured differently, which could lead to different results and interpretations. Finally, the protocols or method of treatment and the results obtained in each of the studies are quite heterogeneous when compared.

This present study carries implications for clinical practice, supporting the development and expansion of healthcare policies regarding telemedicine services in dermatology, and the results could be extrapolated to other healthcare services, such as online models focused on the patient [31].

Current clinical practice shows that chronic dermatological conditions often require regular dermatological visits and are generally associated with high costs for medical care systems [31]. The TD model represents an innovative and profitable method that provides follow-up care to patients when applied properly with relatively low-risk conditions [32].

This work is intended to be a tool to promote the creation of TD protocols that favor diagnoses and optimal treatments, ultimately reducing both societal and economic costs and providing dermatological care in our society that is more accessible and of higher quality [36].

## 5. Conclusions

The studies included in this investigation consider that follow-up via teledermatology can be more cost-effective than face-to-face follow up.

Nevertheless, teledermatology can vary from one country to another, according to the healthcare system in the studies analyzed. Therefore, more work remains to be done for the purpose of establishing cost-effectiveness evaluation protocols that measure the main variables more heterogeneously and with larger samples of participants, making it possible to conduct analyses and thus demonstrate the quality of the evidence produced by rigorous systematic reviews.

## Figures and Tables

**Figure 1 ijerph-19-02534-f001:**
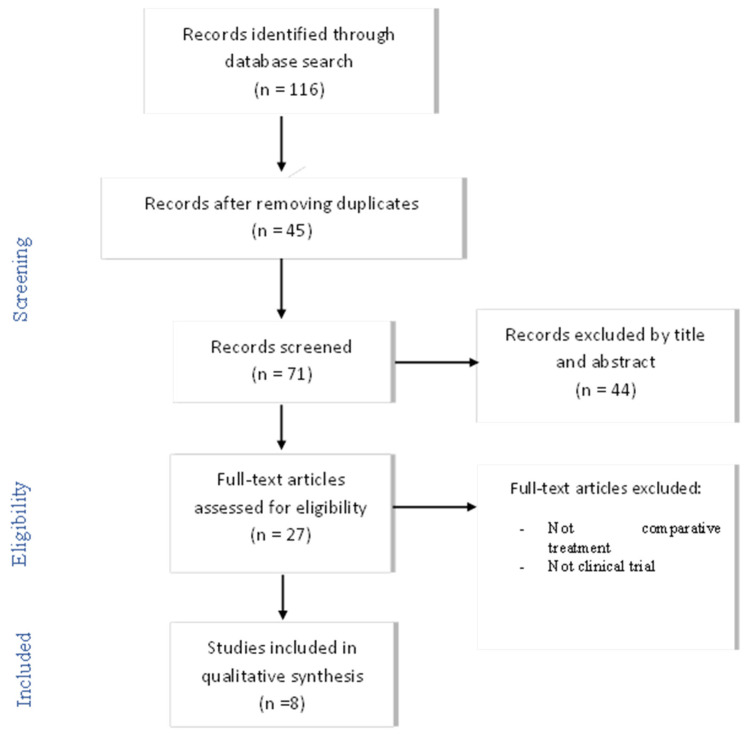
Flowchart of articles selection process. n = number of articles.

**Table 1 ijerph-19-02534-t001:** Databases and search terms.

Databases and Search Terms	Results	Selected Articles
SCOPUS, SCIELO, MEDLINE “Cos-Benefit Analysis” AND “Dermatology” AND “Telemedicine” “Cost-Utility Analysis” AND “Dermatology” AND “Telemedicine” “Economic Evaluation” AND “Dermatology” AND “Telemedicine” “Cost-Effectiveness Analysis” AND “Dermatology” AND “Telemedicine” “Cost- Benefit Analysis” AND “Dermatology” AND “Primary Health Care” “Teledermatology” AND “Remote Consult * AND “Conventional” OR "Face-to-face" OR “Standard” OR “In-person” AND Cost * “Telederm*” AND “Telemed *” AND “Dermatol *” AND “Skin Disease” AND “Telehealth” AND “Dermatol *” AND “Conventional” OR “Standard” OR “Face-to-face” OR “In-Person” OR “Primary Health Care” OR “Remote Consult*” AND “Cost *” AND “Cost-Benefit Analysis” AND “Cost-Utility Analysis” AND “Economic Evaluation” AND “Cost-Effectiveness Analysis”	100	26
PUBMED “Cos-Benefit Analysis” AND “Dermatology” AND “Telemedicine” “Cost-Utility Analysis” AND “Dermatology” AND “Telemedicine” “Economic Evaluation” AND “Dermatology” AND “Telemedicine” “Cost-Effectiveness Analysis” AND “Dermatology” AND “Telemedicine” “Cost- Benefit Analysis” AND “Dermatology” AND “Primary Health Care” “Teledermatology” AND “Remote Consult * AND “Conventional” OR "Face-to-face" OR “Standard” OR “In-Person” AND “Cost * “Telederm *” AND “Telemed *” AND “Dermatol *” AND “Skin Disease” AND “Telehealth” AND “Dermatol *” AND “Conventional” OR “Standard” OR “Face-to-face” OR “In-Person” OR “Primary Health Care” OR “Remote Consult*” AND “Cost *” AND “Cost-Benefit Analysis” AND “Cost-Utility Analysis” AND “Economic Evaluation” AND “Cost-Effectiveness Analysis”	16	1

“*” = Truncation.

**Table 2 ijerph-19-02534-t002:** Description of the main results in the selected articles.

Author, Year Country	Type of Study Participants Diagnosis	Dermatology Follow-Up Method	Main Results
Zakaria, 2020 [24] United States	A retrospective cost-minimization analysis In TD: - 1297 patients did not require in-person consultation - 801 patients required in-person consultation In conventional care: - 646 patients did not require dermatological consultation - 1452 had consultation and follow-up Diagnosis of benign processes, infections and seborrheic dermatitis	Compared dermatology patients within a TD triage system vs. a conventional dermatology care model Asynchronous 1st PC consultation + TD consultation Follow-up: 6 months	Average cost/patient TD: $559.84 Average cost/patient conventional consultation: $699.96 TD reduces costs by $140.12 ($11.01) per patient, representing approximately $441,378 per year.
López-Villegas, 2020 [25] Spain	Inter-level multicentre retrospective study *n* = 7030 patients - 2629 patients required in-person consultation - 4349 patients diagnosed with TD 108 required clarification from PC personnel 2 excluded, software error 3507 diagnosed and treated by dermatologist using the platform 730 received recommendations from PC doctor Diagnoses: non-benign pathologies (basal-cell carcinoma, melanoma, squamous-cell carcinoma, other types of skin cancer) and Benign pathologies (actinic keratosis)	Teledermatology units compared to conventional monitoring at hospitals Asynchronous 1st PC consultation + TD consultation Follow-up: 1 year	Savings of 61.86% of hospital visits There was a cost savings of 31.68% in the TD group (€18.59 TD vs. €27.20 CM) during the follow-up period. The number of CM visits to the hospital reduced by 38.14%. From the patients’ perspective, the costs were lower, and the cost savings was 73.53% (€5.45 TD vs. €20.58 CM). TD is an economic solution in comparison with conventional dermatology, both from the perspective of the public healthcare system and the patient, obtaining clear cost savings The results cannot be extrapolated to countries with different healthcare systems
Vidal-Alaball, 2018 [26] Spain	Observational Study *n* = 5606 patients 1104 patients sent to in-person consultation No specific mention of main diagnoses	A cost–savings analysis comparing teledermatology with traditional dermatology consultations (face-to-face) Asynchronous 1st PC consultation + TD consultation Follow-up: 3 months	TD saved 4502 in-person visits. The use of TD versus face-to-face consultations generates savings of 51,164.00 euros/year (11.40 euros/patient) Societal savings are the most significant Cannot be extrapolated to countries with different healthcare systems
Yang, 2018 [27] United States	A retrospective study of the cases *n* = 700 patients 189 patients required in-person consultation 23 required urgent care 86% of patients were assisted via TD	Analysis compared the cost of each patient case with use of the TD consultation model vs. conventional care Asynchronous 1st PC consultation + TD consultation Follow-up: 5 years	Compared with conventional care, TD had an average expected cost savings of $10.00 to $52.65 per TD consultation Providing access to high-quality care, this program reduced unnecessary in-person dermatology clinical visits as well as urgent care Significantly reduces the costs of dermatological care, increases access and improves patient satisfaction and clinical results
Zarca, 2018 [28] France	Retrospective cohort study *n* = 450 prisoners 82% (368/450) of the patients with tele-expertise had a complete treatment plan The most frequent lesions were acnea and atopic dermatitis	Evaluate the effectiveness and costs of tele-expertise in dermatology Follow-up: 1 year	For 368 patients every year, the average cost is €184 for completed treatment plans distributed as follows: 34% investment, 66% operating cost (30% human resource, 36% software) Tele-expertise was found to be effective, increasing the proportion of patients with completed treatment plans and the overall satisfaction of physicians, at a cost far lower than a dermatologist consultation in a hospital
Datta, 2015 [29] United States	Randomized clinical trial Conventional *n* = 196 patients TD: *n* = 195 patients 1 patient excluded, assignment error Ambulatory skin conditions	To assess the costs and utility of a store-and-forward teledermatology referral process compared with conventional referral process. Asynchronous 1st PC consultation PC + TD consultation Follow-up: 9 months	The TD cost per patient was $30 lower and, from a societal perspective, saved $82 The differences in effectiveness for both groups were not significant Compared with conventional referrals, store-and-forward teledermatology referrals were performed at a comparable cost (VA perspective) or at a lower cost (societal perspective) with no evidence of a difference in utility as measured by the time trade-off method
Os-Medendorp, 2012 [30] Netherlands	Randomized controlled study with economic evaluation *n* = 199 patients with atopic dermatitis CG: *n* = 98 IG *n* = 101	Determine the cost-effectiveness of individualized e-health compared with usual face-to-face care Asynchronous 1st appointment with dermatologist + TD Follow-up: 3 months and after 1 year	Both interventions proved effective regarding quality of life and severity of illness, but there were no significant differences. The difference in total costs between the CG and the IG is €594 per patient during the first year of treatment IG: average 3378 € per patient CG: 3972 € per patient It is possible that telemedicine is more economical because it involves fewer days of work absenteeism; although the results may not be consistent due to the heterogeneity of the interventions Uncertainty analyses revealed that the probability of e-health reducing costs was estimated to be around 73%
Parsi, 2011 [31] United States	Randomized Controlled Trial Cost-effectiveness Analyses *n* = 64 patients with psoriasis.	Compare cost-effectiveness of conventional in-office care with a patient-centered, online model for follow-up treatment Asynchronous. Remote monitoring by the patient (previous training) Follow-up: 24 weeks	Both improved patient quality of life and proved effective: patients gained 23.3 weeks of quality of life with TD and 24.1 with in-person care The online model is equally effective but less expensive than face-to-face: The cost of follow-up care with online visits was 1.7 times less than the cost of in-person visits ($315 vs. $576). TD costs $261.10 less per patient. Users also expressed their preference for TD. TD appears to be an innovative and profitable healthcare model for follow-up and treatment of patients with psoriasis

**Table 3 ijerph-19-02534-t003:** Description of “Cost and Effectiveness” Variables.

Author, Year	Direct Costs	Indirect Costs	Exclusion of Costs	Effectiveness Variables
Zakaria, 2020 [24]	Costs associated with PC, dermatology and TD visits (number of visits) Personal costs: salary and time Technological costs: installation of licensed software, software support, maintenance services, equipment and training for professionals	No reference made to indirect costs	Exclusion of rent costs, clinical supplies, public services and social costs	Does not include effectiveness
López-Villegas, 2020 [25]	Costs from PHS perspective: Number of both conventional and TD visits PC and hospital costs: labour costs and specific costs of TD equipment and materials (Canon EOS camera and dermascopy lens); every 5 years	Patient Cost Perspective: Transport costs, travel time, waiting time and time of visit	Exclusion of structural costs (electricity, telecommunications, construction and maintenance) Exclusion of number of trips and wages lost by companions	Effectiveness was not measured
Vidal-Alaball, 2018 [26]	Number of PC, dermatological and TD visits. Direct costs: cameras, hardware and personnel Equipment costs: iPad Air with WIFI and a mobile phone with 32 GB; every 5 years.	Costs included building maintenance, IT services, gas, electricity, telephone–internet connections and medical insurance Patient costs: Loss of productive time, salary and transport costs (time and fuel)	Exclusion of costs incurred by companions, loss of free time, technical and maintenance costs, training costs for professionals and medical insurance	Effectiveness was not measured
Yang, 2018 [27]	Direct medical costs of health care: average cost of an in-person dermatology visit, an emergency dermatology visit, and the dermatologist for a TD visit	Indirect costs were not included	Separate costs for medications, patient travel, laboratory work and/or imaging or procedures outside of the visits were not included in the analysis	PC doctors were asked about their satisfaction with using TD
Zarca, 2018 [28]	Evaluates image quality but not diagnostic precision Costs of transport, facilities and hospitals Investment costs Operational costs: maintenance and human resources	The proportion of patients with a completed treatment plan for skin lesions, the proportion of technical problems, the quality of the pictures, the investment and operating costs and the satisfaction of the professionals Average cost of a complete treatment plan: investment costs + operational costs	They could not evaluate the diagnostic accuracy of telemedicine; the causes of “failed upload, unanswered requests and unsatisfactory requests”; unable to measure patient satisfaction	Satisfaction survey for doctors.
Datta, 2015 [29]	The cost elements from the VA perspective included: TD intervention or referral; referral and follow-up visits to the dermatology clinic; dermatological medications prescribed; travel cost reimbursement to patients paid by the VA; and dermatology-related hospitalizations	Dermatological care sought outside of the VA system; travel costs in seeking health care if not reimbursed by the VA; and work or productivity loss owing to the patient	They did not include the equipment cost in TD	Time trade-off determines the quality of life one experiences in a given state of health by assessing the equivalence point between living a longer life with the given medical condition vs. a shorter life in perfect health
Os-Medendorp, 2012 [30]	Costs of primary care, e-health service and outpatient clinics Care at the dermatology department Telephone consultation with a dermatologist Combined visit to dermatology nurse and dermatologist Day of hospitalization Number of e-consultations with dermatology nurse Care at other hospital departments Care by the general practitioner	Two modules online of the ‘Health and Labour Questionnaire’ Written diary to gather data regarding days on leave from work Visits with transportation and parking costs to a medical specialist or nurse in a hospital; a general practitioner; days of work absenteeism; hours with loss of productivity during work; hours with loss of productivity in unpaid work	Equipment costs not included	Quality of life, itch intensity and severity of atopic dermatitis *DLQI *IDQOL *VAS Two parts of the online questionnaire “Impact of Chronic Skin Disease on Daily Life” (shortened version)
Parsi, 2011 [31]	Social perspective: Online costs: Medical: clinic and facility fees, morbidity and mortality Non-medical: loss of productivity, cost of patient’s time, equipment (camera, computer) In-office costs: Medical: clinic and facility fees, morbidity and mortality Non-medical: loss of productivity, cost of patient’s time (including travel costs)	Costs of patient’s time: patient wages/minute, duration of visit, cost of patient’s time/visit	Not included	Quality-adjusted life expectancy *HLQOL *QALY *DLQI *QALE *EQ-5D

PC = Primary care; TD: Teledermatology; DLQI = Dermatology Life Quality Index; IDQOL = Infants’ Dermatitis Quality of Life Index; VAS = Visual Analogue Scale; HLQOL = Quality of life measured by the current health condition of an individual; QALY = Quality-adjusted life years; QALE = Quality-adjusted life expectancy; EQ-5D = European Quality of Life 5-Dimension Questionnaire.

**Table 4 ijerph-19-02534-t004:** Methodological Quality of Studies.

	Study Quality										External Validity			Study Bias						Confounding and Selection Bias							Study Power		
Author (Year)	1	2	3	4	5	6	7	8	9	10	11	12	13	14	15	16	17	18	19	20	21	22	23	24	25	26	27	Total	Quality
Zakaria, 2020 [24]	1	1	1	1	0	1	1	1	1	1	1	1	1	1	1	1	1	1	1	1	1	1	0	0	0	1	1	23	Good
López-Villegas, 2020 [25]	1	1	0	1	0	1	0	1	1	1	1	1	1	1	1	0	1	1	1	1	1	1	0	0	0	1	1	21	Good
Vidal-Alaball, 2018 [26]	1	1	0	1	0	1	0	1	0	0	1	1	1	1	1	1	1	1	1	1	1	1	0	0	0	0	0	17	Fair
Yang, 2018 [27]	1	1	1	1	0	1	1	1	1	1	1	1	1	1	1	0	1	1	1	1	1	1	0	0	0	1	1	22	Good
Zarca, 2018 [28]	1	1	1	1	0	1	1	1	1	0	1	1	1	1	1	1	1	1	1	1	1	1	0	0	0	1	0	21	Good
Datta, 2015 [29]	1	1	1	1	0	1	1	1	1	1	1	1	1	1	1	1	1	1	1	1	1	1	1	1	0	1	1	25	Good
Os-Medendorp, 2012 [30]	1	1	1	1	0	1	1	1	1	1	1	1	1	1	1	1	1	1	1	1	1	1	1	1	0	1	1	25	Good
Parsi, 2011 [31]	1	1	0	1	0	1	1	1	0	1	1	1	1	1	1	1	1	1	1	1	1	1	1	1	0	0	1	22	Good

## Data Availability

The datasets used and/or analyzed during the current study are available from the corresponding author upon reasonable request.

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
