# Peer review of "Teledermatology versus Face-to-Face Dermatology: An Analysis of Cost-Effectiveness from Eight Studies from Europe and the United States"

_ijerph, 2022, doi:10.3390/ijerph19052534_

Round 1
Reviewer 1 Report
Reviewer’s comments 2nd February 2022
|
Title could be more precise eg |
||
|
Teledermatology and face to face dermatology. An analysis of cost effectiveness from eight studies from Europe and the USA
The word conventional could be replaced with Face to face as indications are that soon TD will be the conventional method of interaction.
Abstract could mention the countries from which the 8 papers were analysed. The title should also reflect this as suggested
The exact period – start date and end date when the study was carried out should be highlighted.
|
Author Response
All the suggested comments have been included in the manuscript.

Reviewer 2 Report
Please see the attached file.

Author Response
We would like to express our sincere gratitude for all comments and suggestions received. Thank you very much indeed.
